# Multi-Omics Approaches to Study Molecular Mechanisms in *Cannabis sativa*

**DOI:** 10.3390/plants11162182

**Published:** 2022-08-22

**Authors:** Tiziana M. Sirangelo, Richard A. Ludlow, Natasha D. Spadafora

**Affiliations:** 1CREA—Council for Agricultural Research and Agricultural Economy Analysis, Genomics and Bioinformatics Department, 26836 Montanaso Lombardo, Italy; 2School of Biosciences, Cardiff University, Sir Martin Evans Building, Museum Avenue, Cardiff CF10 3AX, UK; 3Department of Chemical, Pharmaceutical and Agricultural Sciences, University of Ferrara, 44121 Ferrara, Italy

**Keywords:** cannabis, genomics, metabolomics, multi-omics, transcriptomics

## Abstract

Cannabis (*Cannabis sativa* L.), also known as hemp, is one of the oldest cultivated crops, grown for both its use in textile and cordage production, and its unique chemical properties. However, due to the legislation regulating cannabis cultivation, it is not a well characterized crop, especially regarding molecular and genetic pathways. Only recently have regulations begun to ease enough to allow more widespread cannabis research, which, coupled with the availability of cannabis genome sequences, is fuelling the interest of the scientific community. In this review, we provide a summary of cannabis molecular resources focusing on the most recent and relevant genomics, transcriptomics and metabolomics approaches and investigations. Multi-omics methods are discussed, with this combined approach being a powerful tool to identify correlations between biological processes and metabolic pathways across diverse omics layers, and to better elucidate the relationships between cannabis sub-species. The correlations between genotypes and phenotypes, as well as novel metabolites with therapeutic potential are also explored in the context of cannabis breeding programs. However, further studies are needed to fully elucidate the complex metabolomic matrix of this crop. For this reason, some key points for future research activities are discussed, relying on multi-omics approaches.

## 1. Introduction

Cannabis (*Cannabis sativa* L.) is an herbaceous species originating from Central Asia and is distributed to lesser extent all over the world, growing in wide ranging habitats and climatic conditions [1].

Cannabis is considered one of the oldest cultivated multipurpose crops. In fact, it can be classified as fibre crop (hemp), with a long history of rope and textile making, thanks to its cellulosic and woody fibers, and also a drug crop (medicinal cannabis) since it is used for therapeutic purposes [2]. Hemp contains less than 0.3% of tetrahydrocannabinol (THC), whereas medicinal cannabis contains a greater amount of this metabolite, generally accounting for up to 5% of the dry weight [3]. THC, along with cannabidiol (CBD), are the most important secondary metabolites produced by cannabis and, among the ~130 secondary metabolites identified [4,5], they are the predominant focus of breeding programs and pharmaceutical industries [2,6]. Abbreviations used throughout the manuscript are listed in Table 1.

Despite their similar chemical structures, these two metabolites do not have the same effects on the human body [2]. THC is psychoactive and induces a sense of euphoria, while CBD is not, but instead has therapeutic uses in reducing anxiety and depression [2].

In spite of the small size of the genus, the exact number of cannabis species is still not well defined. According to scientific studies, there are three cannabis species with distinct phenotypic differences, namely *C. sativa* L., *C. indica*
*Lam* (Lamarck), and *C. ruderalis* [7,8]. However, the majority of classifications performed to date evidence the existence of *C. sativa* and *C. indica* only. Specific crosses between these two species are referred to as hybrids and have highly variable phenotypes, showing intermediate features to the parents [9]. However, these common distinctions are not representative of the evolutionary relationships [10]. There is a relevant phenotypic variation among these cannabis species, especially involving cannabinoids [11,12] and terpenoids levels [13,14], as well as differences among genotypes [7,9,15], and morphologic features like flowering time, and branch and internode length [1]. Morphologically, *C. sativa* plants are tall, have less dense buds, narrow leaves, and produce high levels of THC. Conversely, *C. indica Lam* plants are short with denser buds and broader leaves, and they synthetise high levels of both THC and CBD [1,9,11].

Within a given cannabis species, cultivars are categorised into groups based on their chemotype, from I to V, according to the number and ratio of main cannabinoids [16]. These compound profiles can be employed both as quality markers and fingerprints for cannabis standardization.

Cannabis growth can be divided into four distinct phases: germination, seedling, vegetative and flowering stage. Each phase is characterised by its own photoperiod, environmental and nutritional needs [17]. Cannabis germinates, reaches maturity, reproduces and dies in one year in the wild. The flowers are unisexual, and therefore male and female individuals are distinct. However, hermaphrodites have often been documented [17,18]. In general, males and females are not identified until the second week of the bloom cycle. Since only female inflorescences are used to produce extracts, once a male plant is identified, it is generally discarded. The vegetative phase is characterised by the greatest increase in biomass and overall growth. During this phase, roots extend considerably, leaves start growing and expanding to increase the photosynthetic area, and transpiration rises, so water intake increases as well. The reproductive phase of cannabis development involves massive hormonal changes induced by photoperiod and this can be enhanced by an increase in red and far-red wavelengths of light [17,18]. In this phase, the first major increase in cannabinoid levels occurs in female inflorescences [17,18].

Due to the legislation regulating cannabis cultivation, the cannabinoid biosynthetic pathway has not been characterized in detail, especially from a molecular and genetic perspective [19]. Conversely, many other major crop species have already been widely studied, especially after the development of next generation sequencing (NGS) technologies [19,20]. The recent relaxation in legislation [21,22], as well as the availability of the cannabis genomic sequence, consisting of a complex genome, containing 843 Mb and 818 Mb for male and female plants, respectively [23], is now facilitating research on this crop. The diploid genome consisting of nine autosomes and a pair of sex chromosomes [24,25], is highly heterozygous, and contains many repetitive elements (~70%) [5,26,27]. Despite the presence of distinct sex chromosomes, some external factors, such as a shorter photoperiod, a lower temperature, and the application of chemicals, such as ethylene inhibitors, on leaves, can enhance pollen production in female flowers, resulting in ‘feminised seeds’ [28]. This technique has been often used in cannabis breeding to generate target populations to investigate key phytochemical and qualitative traits.

The presence of many lipophilic cannabinoids and terpenoids in cannabis, along with numerous other primary and secondary metabolites, adds complexity to metabolic analyses [20]. Therefore, a detailed analysis at both the chemical and molecular level is extremely challenging, requiring the use of innovative and advanced approaches. NGS technologies have revolutionized plant biology and have been applied widely to non-model systems with relatively low costs. This is presently facilitating the application of combined approaches, based on the relationship between genomic, transcriptomic and metabolomic methods, exploiting the inter-related data sets [29]. Multi-omics has already been used successfully to identify correlations between different biological components and metabolic pathways in several other crops [30]. These approaches have facilitated a comprehensive genetic and metabolic mapping of cannabis, allowing an accurate characterisation and further exploration of: (i) the relationships between sub-species, (ii) the relationship between genotypes and phenotypes, and (iii) novel metabolites/biomarkers in cannabis breeding programs.

In this work, we review the available genomic resources, and recent genomic, transcriptomic and metabolomic tools for improving cannabis knowledge, with a focus on multi-omics approaches. With the term ‘multi-omics’, we refer to any approach applying the principles of different -omics sciences to analyse a given issue, not necessary based on an initial integration of different omics data sets. The integration and concurrent analysis, which allows a more comprehensive and powerful interpretation of multi-omics data, is complex and requires advanced bioinformatic pipelines [31]. However, discoveries have been made through more simplistic analysis of each-omic dataset individually and then comparing the results to look for correlations. Research perspectives are examined for both these strategies, and suggestions made for possible future studies.

## 2. Studying the Metabolomic Profile of Cannabis

### 2.1. Key Metabolites: An Overview

Cannabis is a polymorphic plant species producing a diverse profile of bioactive metabolites which have unique chemical structures and physiochemical properties [20]. Among them, the main compound class are the cannabinoids, accounting for ~20% of the total secondary metabolites in cannabis, and terpenoids are also highly abundant, of which isoprenes, monoterpenes, and sesquiterpenes are predominant [32].

Cannabinoids are primarily synthesized in the glandular trichomes of female flowers, while trichomes of male flowers are generally very low in cannabinoids [33]. Cannabis trichomes are classified as stalked, sessile, or bulbous, where bulbous trichomes produce limited cannabinoids compared to the other types [34]. Trichomes contain resin storage cells and, during the flower and seed maturation stage, the composition of cannabinoids within the resin changes, reaching the highest level at flower maturity [33]. The concentration of cannabinoids increases in warmer temperatures but is negatively correlated with the mineral content of soil [35]. Cannabinoid yield is also affected by UV-radiation and an increase was observed in cannabis flowers after UV-B-induced stress [36].

Figure 1 illustrates the main steps of THC and CBD biosynthesis. THC and CBD are synthesised from two distinct metabolic pathways: the polyketide and the methylerythritol phosphate (MEP), producing olivetolic acid (OA) and geranyl diphosphate (GPP), respectively [5,37]. Specifically, OA and GPP synthesize the cannabigerolic acid (CBGA), containing a pentyl side chain, which produces the acidic precursors of THC (THCA) and CBD (CBDA) [38]. The cannabichromenic acid (CBCA) is also produced [39]. Synthesis of THCA, CBDA, and CBCA proceeds through the appropriate oxidocyclases: THCA synthase, CBDA synthase, and CBCA synthase, respectively.

These acidic cannabinoids are thermally unstable and can be decarboxylated when exposed to light or heat via smoking [40]. Terpenoids are produced by dimethylallyl diphosphate (DMAPP) and isopentenyl diphosphate (IPP) metabolic pathways, which share a GPP precursor with cannabinoids [41].

However, the biosynthesis of cannabinoids and terpenes is still far from being fully understood at the molecular level [34]. Further and innovative investigations are crucial for many upcoming medicinal cannabis applications, where novel bioactive compounds or less abundant cannabinoids may be of great interest [42].

### 2.2. Cannabis Metabolite Profiling Techniques

The chemical composition of cannabis is extremely important. In fact, it can be unique for each cultivar and the metabolic fingerprint is fundamental to exploring the differences among them [43].

A variety of techniques have been employed to extract and analyse compounds from cannabis, and different methods are used depending on whether the aim is to investigate cannabinoids or terpenes [44].

The most popular platforms used to analyse cannabinoids are gas chromatography (GC) and liquid chromatography (LC), coupled with mass spectrometry (MS) [44,45]. LC is used for the analysis of non-volatile and thermally labile compounds, while GC allows the analysis of thermally stable molecules and often derivatization agents are used to aid this process. GC-MS usually uses electron ionization (EI) to fragment the analytes in a consistent way, whereas LC-MS generates ions with less diagnostic fragmentation information. Considering the complex metabolomic matrix of cannabinoids, working on less abundant, more novel cannabinoids is challenging [44]. Cannabis testing laboratories often prefer to use LC for cannabinoid chemical analysis, due to simpler sample preparation steps. For instance, the derivatization and decarboxylation of related precursory molecules, which are necessary for GC based methods, can be skipped with LC [44].

GC-MS was employed in an interesting study recently [46] where the effects of natural and artificial lighting on cannabinoid metabolism were analyzed. Specifically, treatments of cannabis crops with 3 different light spectra, high-pressure sodium (HPS), AP673L (LED), and NS1 (LED) were investigated [46]. Results explored how these treatments affected cannabis morphology and its CBG, CBD and THC content, but they only had a minor impact on the overall yield. Furthermore, LED lights resulted in higher amounts of plant growth and improved the cannabinoid profile, when compared to HPS lights. Plants grown under LED light conditions had a boosted THC and CBD concentrations, and cannabis cultivars having an elevated THC yield also exhibited a higher photosynthetic capacity. This suggests that different cannabis chemovars may be optimally cultivated under different light intensities. The potential of LED lighting in the cannabis sector has been further investigated, but there is a lack of tangible evidence on how light quality and light source affects extract quality and yield [36]. Furthermore, a recent study based on gas chromatography electron impact mass spectrometry (GC-EI-MS) investigated the lipids extracted from seeds of *C. sativa* and identified over 40 cannabinoids. Indeed, 16 of which had never been detected before, and some were hoped to have future medicinal potential [47].

High Performance Liquid Chromatography (HPLC) has also been widely applied in the study of cannabis metabolites, with it being faster, more sensitive and efficient compared to LC. It was used in a study in which a set of cannabis varieties, representative of all chemotypes, were analysed and compared [48]. The total yield of the major cannabinoids CBD, CBG and THC were measured in female monoecious hemp inflorescences. The varieties with the highest CBD content were ‘CS’ and ‘Carmagnola’, while the lowest amount of CBD was found in ‘Santhica 27’. Conversely, ‘Bernabeo’ genotype showed the highest value of CBG and, as expected, the THC content of the medical varieties, like ‘CINBOL’ were very high. Another study based on HPLC showed that the content of cannabinoids is highly influenced by the cultivar and the plant growth stage [49]. Specifically, the investigation was focused on a set of industrial hemp cultivars, and the results demonstrated that, although some of them, e.g., ‘Futura75’, ‘Fédora17’, ‘Félina32’, and ‘Ferimon’, are mainly cultivated for fibre and seed production, they can also be used for cannabinoids extraction. Furthermore, these cultivars showed a maximum CBDA yield when the seed completed its maturation.

Besides cannabinoids, terpenes are another important compound class in cannabis. They are usually analysed by GC, coupled with various detectors, such as Flame Ionization Detection (FID), which is the most commonly applied due to its low cost and ease of use. A study based on this platform allowed the classification of 13 cannabis cultivars based on their terpenoid profile [50]. Specifically, results highlighted how some cultivars fit into one or more specific chemotypes, whereas in other cultivars this association is not so clear-cut [50]. More recently, by taking a combined analysis approach using both a LC-diode array detector (LC-DAD) and GC-FID technologies, further classifications based on cannabinoid and terpenoid contents were proposed [43]. The results confirmed how the chemical composition is specific for each cultivar.

Depending on analytical aims, other extraction protocols can be applied to analyse cannabis metabolites, including supercritical fluid extraction (SFE), an extraction method using supercritical CO_2_ which is less expensive and more effective compared to chemical solvents, and represents a valid alternative to classic extraction systems [51]. For instance, sequential SFE and solid phase extraction (SPE) processes allowed THC to be extracted at a purity level suitable for quality control, where SPE was used as a purification technique for THC [52].

Furthermore, triple-quadrupole mass spectrometry techniques, known as QQQ and based on tandem MS, in which the first and third quadrupoles act as mass filters and the second fragments the chemical component, have also been applied to cannabis [53]. Due to its improved selectivity and sensitivity, this method was proven to be highly effective in quantitative cannabis metabolite analyses [53]. Therefore, it would also be possible to use this method to evaluate the abundance of cannabinoids in different parts of the crop, which may not be detected with other approaches given their low abundance [54].

Another versatile method increasingly used for the detection of analytes in complex matrices is the nuclear magnetic resonance spectroscopy (NMR) [55]. This technique is characterized by a low sensitivity compared to MS, even if it provides more reliable metabolite structure and does not require destructive sample preparation [56]. Furthermore, NMR allows simultaneous identification of multiple analytes. Despite these potential advantages, NMR has only been rarely applied in the detection of cannabinoids: as far as we know, it was mainly used to the authentication of hemp varieties [57,58].

In cannabis, due to the complex metabolome, the combination of several analytical methods usually gives the most comprehensive picture [2]. For instance, LC/QQQ/MS and NMR metabolomics analyses revealed the presence of several cannabinoids detected in extracts of cells of capitate-sessile and capitate-stalked trichomes as well [59]. Extracting and analysing the chemical profile of specific trichome lines holds great potential for use in future multi-omics experiments, as transcriptomic analyses could also be performed on these specific cell types [33]. The relationship between compound profile within these cells and their relative gene expression of cannabinoid biosynthetic genes could yield exciting new insights.

Another powerful technique used for the analysis of metabolites in cannabis is High-resolution mass spectrometry (HRMS), this technique enables a more precise identification of compounds with the same nominal mass due to the improvement in the calculated mass to charge ratio (*m*/*z*) to several decimal places compared to conventional MS. The use of this technique was found to have a great potential in the definition of cannabis chemovars [60]. Indeed, a recent study employed this method [61]: by using data from 20 varieties of *C. sativa* and a combined LC-HRMS platform, metabolites were mapped and annotated, and cannabis characteristic markers identified. The results of this approach were compared with those based only on major cannabinoid quantification, and it was found that minor compounds were highly predictive markers for differentiating cannabis varieties. Furthermore, these findings may be combined with other data coming from multi-level omics investigations, confirming the applicability and the potential of metabolomics in the understanding of cannabis metabolism regulation mechanisms.

## 3. Genomic and Transcriptomic Cannabis Profiles

Initial attempts to assemble the complex cannabis genome relied on the use of short-read sequencing technologies, but only recently have third-generation long-read sequencing technologies, such as Single-Molecule Real-Time (SMRT) sequencing (PacBio) and Oxford Nanopore Technologies (MinION) [25], led to an improved contiguity of reference sequences and correctly assembled ambiguous, highly repeated regions [25,62].

These advances resulted in the creation of four assemblies related to different cultivars: ‘Purple Kush’ (‘PK’, a drug type Cannabis), ‘Finola’ (‘FN’; a fiber type Cannabis), ‘Jamaican Lion’ (‘JL’; a wild accession) and ‘CBDRx’ (‘cs10’; with high CBD content) [27,63]. Comparisons of the transcriptome of ‘PK’ with that of the hemp cultivar ‘FN’ revealed that many genes encoding proteins involved in cannabinoid pathways are more highly expressed in ‘PK’ than in ‘FN’. Subsequently, these reference assemblies were annotated with full-length male and female mRNA sequencing to provide better information about isoforms complexity, genes and Y chromosome identification [25]. To date, the ‘cs10’ genome sequence assembly is considered to be the most complete and contiguous genome and is broadly used as the reference genome for cannabis [64].

The presence of copy number variations (CNVs) in cannabinoid synthases have been demonstrated in several cannabis genome studies [25,63] while the relation between cannabinoid synthase CNVs and cannabinoid content is still not clear [19]. Furthermore, highly similar loci are not adequately differentiated in short read sequencing approaches [19,65].

The availability of a sufficiently complete genome is crucial for the understanding of the cannabinoid pathways through a better knowledge of the underlying genes. Cannabinoid biosynthesis was investigated at molecular level, and several genes involved in this pathway were identified [48]. Furthermore, it was found that each gene consists of a single exon, with THCA synthase (THCAS) and CBCA synthase (CBCAS) sharing over 90% homology at amino acid level and over 80% homology to CBDAS [62,66].

The availability of these more complete genomic resources allowed the identification of sex chromosomes and of approximately 3500 gender-specific genes in the cannabis genome [25,67]. THCA and CBDA were found to be mainly produced in the inflorescences of female cannabis crops [24,25]. Thus, detecting male and female plants at early growth stages can increase yield and help design more specific cannabis breeding programs.

The association between THCA and CBDA synthase sequences and Quantitative Trait Loci (QTL) has been also reported [8,38]. An early study, using bi-parental mapping populations coming from a cross between hemp and drug cannabis, identified QTL regulating biochemical traits. Results suggested that THCA and CBDA synthase sequences are associated to a single multiple linked QTL [38]. Another study used a set of over 20 highly informative single nucleotide polymorphisms (SNP) markers related to cannabinoid and terpenoid expression to assess phylogenetic relationship, population genetics, and correlation with cannabis metabolites, demonstrating the utility of this method for efficient genotyping activities [8].

Other quite recent studies have been carried out, based on Genome Wide Association Studies (GWAS) in order to investigate novel cannabis genetic variants responsible for cannabis complex traits. For instance, candidate cannabinoid pathway genes have been identified, focusing on the alkyl side chain group whose genetic basis are mostly unknown although is a critical feature behind health properties [68]. These findings confirmed a previously known locus involved in cannabinoid synthesis pathway and other loci associated to chemotype variability [62], revealing 22 variants in a β-keto acyl carrier protein (ACP) reductase (BKR). It is worth noticing that genetic improvement of the alkyl side chain could help the development of new chemical chemotypes for pharmaceutical use. Furthermore, a GWAS approach has been applied to study the genetic architecture of flowering time and sex determination in hemp by using a panel of over 100 hemp accessions and a large set of SNP markers [69]. Several key genes and transcription factors involved in regulating phytohormones levels, like gibberellic acid, were identified in sex determination loci. These QTLs were proved to be responsible for the development of male flowers in female plants, being behind sex determination in monecious plants and its stability over time [69].

However, despite the advancement of the latest cannabis resources, the understanding of the genetic variation underlying complex agronomic traits of interest is still limited [19]. Although other recent investigations have been carried out, the development of efficient NGS tools and the construction of high-density genetic cannabis maps are necessary to improve the QTL mapping quality [62,63].

Genomic selection (GS) methods, which rely on genome-wide marker information to forecast the breeding impact of genotypes, could be a relevant approach to reach this aim. GS has been recently used in breeding other crop species, including rice and canola [70,71]. In both investigations, a multi-omics approach was applied to enhance agronomically important breeding traits including yield, grain weight and hybrid performance, underlining the advantages of combining omics datasets for GS analysis. In rice, the genomics predictions using genomic and metabolomic datasets showed better results than single omics approaches [70]. In canola, both SNPs and transcripts resulted reliable to predict hybrid performance using the most effective genomic unbiased prediction models. Compared to models just relying on pure genetic markers, those taking into account transcriptome data seem to be related to a significantly higher prediction accuracy, suggesting that transcripts contain relevant information beyond just genomic data [71]. The overall results reached to date are promising and open new perspectives for the genetic enhancement of complex traits regulated by a large number of genes. In the future, when further information become available and statistical models and phenotyping accuracy improve, these findings could also be applied to cannabis traits.

Regarding terpene synthase genes, the CsTPS family in the ‘JL’ reference genomes was characterised [72]. Copy number gains in *CsTPS17* were observed in several cannabis cultivars, and *CsTPS17* was identified as potentially involved in myrcene or limonene synthase [25]. In the same study, a copy number analysis in the ‘JL’ genome revealed a unique amplification of Gibberellic Acid Insensitive genes (GAI), which are known to be involved in plant growth. However, further investigations are required to better understand the contribution to yield of these genes.

Pathogen response genes, like those belonging to the Mildew Locus O (MLO) family, correlated with resistance to powdery mildew (PM), and the Thaumatin-Like Protein (TPLs) family, correlated with a wide range of pathogen resistance traits in plants including cannabis, were studied [25]. Specifically, the analysis confirmed extensive CNVs in cannabinoid synthesis and over 80 genes associated with resistance to *Golovinomyces chicoracearum*. Results also showed that plants with low THCA concentrations have a lower resistance to this pathogen. The antifungal response activity (against *Fusarium oxysposum*) of *CsTLP1* was confirmed as well [25].

Gene expression investigations have been crucial for a better understanding of cannabinoid metabolic pathways [73]. A comparison of the transcriptomes of drug and fiber cannabis revealed that expression of the genes involved in the cannabinoid pathway is enhanced in drug cannabis [23], confirming that positive transcriptional regulators of the cannabinoid biosynthetic genes are more active in this variety. Furthermore, it was found that cannabis has more than 1220 transcription factors classified into families, such as MYB, bHLH, and AP2/ERF, which is considerable, but still far less than Arabidopsis, rice and maize [74]. A gland-specific transcription factor, *HlWRKY1*, controlling prenylated flavonoid and bitter acid biosynthesis in *Humulus lupulus*, a species closely related to cannabis, was detected [75]. Only recently THCA synthase promoters controlling expression exclusively in the trichomes were more thoroughly investigated [76]. Specifically, the *CsAP2L1* (AP2-LIKE) and *CsMYB1* (MYB) transcription factors were identified and the existence of a *CsWRKY1* (WRKY) was confirmed. Results suggest that *CsAP2L1* is a transcriptional activator, while *CsMYB1* and *CsWRKY1* are repressors. However, the understanding of transcriptional regulators that control THCA synthase expression and other cannabis metabolic pathways is still limited.

Further transcriptomic experiments were carried out on both fiber and medicinal cannabis crops [24,77]. Fiber cultivars were studied at different developmental stages, using samples from several stem regions, coming from the top, middle and bottom internodes of hemp stems [77]. Cell wall changes were correlated to RNA-Seq data and results showed that the major changes in fibers and gene expression occurred at the internodal regions and that each region of the stem presents a different gene expression profile. The gene ontology enrichment analysis underlined that genes related to the top region belonged to the DNA replication and cell cycle ontologies, the middle region was characterized by processes related to secondary cell wall biogenesis, while the bottom region was dominated by genes involved into phytohormone, as well as in secondary metabolic processes. Furthermore, immature stem tissue was characterized by photosynthesis related genes, along with others involved in the biosynthesis of specific secondary metabolites, mainly indole-containing compounds and oligolignols. Conversely, older, more mature internodes showed higher transcription levels of genes related to phytohormone production, as well as those involved in the lignification process.

In medicinal cannabis, genetic expression analysis in trichomes and leaf tissues facilitated the identification of many enzymes involved in the metabolic pathway of THCA and CBDA. In particular, a comprehensive transcriptome study using cannabis root, shoot, and flower was carried out [24]. There, genes involved in terpene and cannabinoid synthesis were detected and found to have high expression levels in trichomes. The results of this in-depth transcriptomic study represent a significant resource for future cannabis research. However, sparse information is still available today about the expression of genes associated with the synthesis of less abundant cannabinoids [47].

In the last few years, high-quality reference transcriptomes of two cultivars of Cannabis, a high THC cultivar and a CBD plus THC cultivar were assembled [65]. Each transcriptome contained over 20,000 protein-encoding transcripts. Transcripts for the cannabinoid pathway and related enzymes showed full-length open reading frames (ORFs) that align with the genomes of the ‘PK’ and ‘FN’ cultivars. Furthermore, two transcripts for OA were found to map to distinct locations on the ‘PK’ genome, suggesting that genes involved in OA biosynthesis are expressed in several cultivars.

Taking into account recent advances, transcriptomic studies have the potential to address one of the most crucial agricultural issues in many countries: the soil salinity, whose effects affect more than 800 million hectares worldwide [78]. Targeting breeding techniques to improve cannabis’s tolerance of saline and sodic soils is important to ensure high yields and maintain quality traits. A transcriptome of a saline-alkaline resistant cannabis, grown under NaHCO_3_ stress was investigated [79]. An RNA-Seq approach and weighted gene co-expression network analysis (WGCNA) were used to investigate the gene expression profiles and the results showed that some pathways, related to phenylpropanoid and sucrose, nitrogen, and amino acids biosynthesis, may be correlated to the response of cannabis under NaHCO_3_ stress. In the same year, key cannabis salt stress response genes were investigated by comparative transcriptome analyses of contrasting cannabis varieties, namely the W20 and K94 cultivars [80]. Over 80 differentially expressed genes (DEGs) which overlapped in the two cannabis varieties were identified, with more of these being up regulated than down regulated. Furthermore, results underlined how salt stress can induce increases in lipid peroxidation and reactive oxygen species (ROS) in cannabis and upregulate the expression of antioxidant genes as a response to ionic toxicity, as happens in other plants [81]. These DEGs represent potential targets for modern breeding techniques to adapt cannabis to grow efficiently on sodic soils, whilst still maintaining high chemical quality.

## 4. Multi-Omics Approaches in Cannabis

NGS technologies have revolutionized plant biology and elucidated molecular processes in crops of therapeutic importance [82]. Similarly, combined methods, based on the relationship among genomic, transcriptomic and metabolomic approaches, have been developed for a wide set of crop species, finding synergies among these heterogeneous data.

Indeed, multi-omics approaches have been employed in the study of cannabis, such as a combined metabolomic and transcriptomic approach used to examine mature inflorescences [83]. In this study, the plant chemotype was assessed, and qRT-PCR used to quantify transcript levels of cannabinoid synthesis genes, specifically looking into CBDA and THCA synthesis. These sequences were also compared with other existing sequences obtained by RNA-Seq and led to the identification of several SNPs that correlated to the cannabinoid composition of the inflorescence. The results suggested that these variations can have a functional significance, as well as at least partially explain the existence of different cannabis chemotypes [83].

In a subsequent investigation [15], in order to better understand the diversity and the major cannabis ancestry, the first cannabis genome study [23] was further expanded, by re-mapping sequence reads through whole genome shotgun (WGS) sequencing and genotype-by-sequencing (GBS) approaches. Concurrently, a chemotype analysis was performed by HPLC, and over 350 cannabinoids and terpenoids were found. Overall findings showed that, despite several hybridizations, significant ancestral insights related to modern cannabis varieties were found in the genomic results. Particularly, among the studied ~350 Cannabis varieties, the existence of at least three principal diversity groups with European hemp varieties more closely related to narrow leaflet drug-types than to broad leaflet drug-types was demonstrated [15].

Trichomes are a crucial aspect of medicinal cannabis varieties, being the primary storage organ for cannabinoids. Investigations have focused on the changes occurring during flower development, the main phase of metabolite accumulation, using metabolomics and transcriptomics in parallel [33]. SPME, GC-MS, and LC-MS techniques were applied for the quantitative analysis of terpenes and cannabinoids. RNA-Seq gene co-expression networks were developed, and the Pearson’s correlation coefficient was used to quantify the similarity. Results showed how medicinal cannabis trichomes undergo changes to their morphology and metabolite profile during flower development. Furthermore, they demonstrated a high expression of cannabinoid and terpene related genes, as well as the presence of numerous uncharacterized highly co-expressed genes involved in CBDA synthase, suggesting that the glandular trichomes on cannabis flowers are strongly dedicated to cannabinoid and terpene production.

In the same year, nine cannabis cultivars were selected according to several characteristics, such as colour or smell, the glandular trichomes were isolated from each of the cultivars, and their metabolome and transcriptome were analysed and compared by WGCNA [43]. The analysis revealed genes involved in the biosynthesis of both cannabinoids and terpenoids. Furthermore, in addition to the previously characterized terpene synthase genes, e.g., *CsTPS14CT* [(-)-limonene synthase] and *CsTPS15CT* (β-myrcene synthase), other genes, including *CsTPS18VF* and *CsTPS19BL* (nerolidol/linalool synthases), *CsTPS16CC* (germacrene B synthase), and *CsTPS20CT* (hedycaryol synthase), were investigated from a functional point of view, by identifying the synthase pathway in which they were involved. The cannabis terpene synthase gene family is complex, containing at least 55 members. Results demonstrated that subsets of this family are expressed in all plant tissues, including a set of root specific monoterpene synthases [72]. In the light of the role played by volatile terpenes in plant interaction with the soil biome, findings related to the root genes group belonging to CsTPS family, could be the subject of further investigations being particularly interesting for their agronomic impact. For instance, in maize, β-caryophyllene is released from roots under attack of worms [84].

Terpene profiles of cannabis were recently characterized in detail as part of a multi-omic study also focusing on the transcriptome of trichomes [85]. Sequence analysis was performed on a set of cultivars assembled onto the reference genome of ‘PK’ and this revealed about 30 different CsTPS genes, as well as expression variations of genes involved in terpenoid and cannabinoid pathways between the different cultivars. This study, using a new annotation of the ‘PK’ genome, was able to identify ~20 CsTPS gene models, as well as additional genes involved in isoprenoid and cannabinoid biosynthesis. Despite these recent discoveries, the relationship between terpene and cannabinoid biosynthesis to date has not been fully described.

As the relations between cannabinoid synthase CNVs and cannabinoid content are still not clear [19], some recent studies focussed efforts on this issue. A multi-omics approach, based on metabolomics, genomics and transcriptomics, was employed to study the relationship between cannabinoid synthase CNVs and cannabinoid profile [86]. Cannabinoid profiles of 69 Cannabis cultivars of different lineages were assessed using HPLC. Two de novo cannabis genome assemblies and an additional whole genome shotgun data set from a diversity of cultivars were used, and an RNA-Seq method was applied for the transcriptome analysis. Results confirmed that genes involved in the cannabinoids pathway showed several CNVs and were differentially expressed between the cultivars. Furthermore, new insights about a positive correlation between the accumulation of specific cannabinoids, including THCA, and the copy number of certain synthase paralogs were provided.

The sex chromosome evolution in cannabis was also investigated using multi-omics approaches [67]. An RNA-sequencing pipeline was developed using Complete Genomic (CG) sequencing which, unlike other DNA sequencing techniques, allowed sequences with a gapped read structure to be aligned. There, a further genotyping analysis was performed using the cannabis genome developed in 2011 [23] and only SNPs supported by at least three reads were considered. Results identified over 500 sex-linked genes and showed that old plant sex chromosomes can have large non-recombining regions. Particularly, the X-specific cannabis region was found larger compared to other plant systems. Interestingly, the age estimated for the sex chromosomes ranged from ~12 myr to ~29 myr old, which according to the literature, is among the oldest in plants in which the age of sex chromosomes was investigated by sequencing data [87]. Moreover, it was found that the Y gene loss is about 70%, which is much higher than other species, where this loss has been estimated to be ~40% [88]. Cannabis sex-linked genes were also investigated in a subsequent study [25]. Male and female cannabis genomes (cv ‘JL’) were sequenced with Pac Bio, utilizing long-read single molecule sequencing. These assemblies were further annotated with an Isoform sequencing (Iso-Seq) approach, providing further information about isoform complexity. Genes of interest on the Y chromosome, playing a significant role in sex determination and trichome development, were discussed, among them: *FT* (Flowering Locus T), *FY* (Flowering Time control protein), *PIN2* (Auxin efflux carrier component 2), *CRL5* (AP2-like ethylene responsive transcription factor), and *TBL6* (Protein trichome birefringence-like 6).

The genetic controls of complex agronomic and biochemical traits in cannabis is still poorly understood [19], but research has been increasing in this area lately, also exploring how cannabinoid synthase genes influence the THC:CBD ratio and the overall abundance of cannabinoids. Some recent studies are starting to build a picture of these processes. This includes an investigation into two phenotypically distinct hemp cultivars (‘Carmagnola’ and ‘USO31’) [89], where GC-MS results suggested that in cv. ‘USO31’, olivetol synthase is less active compared to cv. ‘Carmagnola’, and this could explain why ‘USO31’ contains a lower concentration of cannabinoids. Using WGS, around 70 QTL were mapped and related to variation in agronomic and biochemical traits. Results showed that differences between ‘Carmagnola’ and ‘USO31’ are mainly controlled by a small number of loci, since most of the QTLs are colocalized. Another study addressing this issue was carried out by [63] applying metabolomic, genomic and transcriptomic methods to explore the characterisation of ‘CBDRx’, a high-CBD cultivar. Cannabinoid analysis was performed with HPLC, a chromosome-resolved reference genome was generated with the MinION ONT platform, and QTL mapping was performed by a high-resolution linkage map from two populations involving ‘Carmen’ hemp crossed with ‘Skunk#1’ marijuana. RNA-Seq libraries were aligned to the reference and assembled into transcripts. Cannabinoid synthase paralogs are arranged in tandem arrays on chromosome 7. Although ‘CBDRx’ is predominantly of marijuana ancestry, it was found that its genome includes a CBDAS introgressed from hemp and lacks a complete sequence for THCAS. Results underlined that variation among cannabinoid synthase loci can affect the THC:CBD ratio, and that variability in overall cannabinoid content among cultivars might also be correlated with chromosomes other than 7 [63]. Furthermore, two candidate genes possibly associated to cannabinoid QTLs were detected: the gene coding *Acyl-activating enzyme 1* (*AAE1*), an enzyme of the hexanoate (a precursor for cannabinoid biosynthesis) pathway [90], and the gene coding for *4-hydroxy-3-methylbut-2-enyl diphosphate reductase* (*HDR*), the last enzyme in the MEP pathway. Despite the breadth of the results to date, further investigations are necessary in order to completely elucidate the biological controls and mechanisms behind the enhancement of cannabinoid expression.

## 5. Conclusions

The development of NGS technologies and recent advances in biotechnology has made it possible to greatly improve our understanding of the genetics and metabolomics that underpin trait differences in cannabis cultivars.

Table 2 shows which omics technology was applied to study cannabis, and highlights how many omics studies are focused on metabolomics, which appears to be the most prevalent approach.

The table also reiterates that very few QTL investigations on cannabis have been completed to date. However, there is a clear increase in transcriptomic studies based on RNA-Seq in recent years, which is a result of the availability of new and better cannabis reference genomes and lower costs associated with creating these datasets [25,65].

NGS has also facilitated a greater uptake of multi-omics approaches, showed in Table 3. The results of the investigations based on these technologies have elucidated more about the complex cannabis metabolomic matrix, and some genes involved in cannabinoid biosynthesis have been identified. A relevant example is the improvement in the characterization of the complex CsTPS gene family, and a better definition of the role of some key genes belonging to this family [43,72,85].

Significant advances in understanding the molecular mechanisms of cannabis have been made in recent years, despite the challenges implementing this interdisciplinary approach. To date, a complete integration of heterogeneous multi-omics data is still extremely challenging and tools to manage and integrate data coming from different omic layers have still to be developed. In this context, the role of publicly available network resources, allowing the access to genomics, genetics, transcriptomics, and metabolomics cannabis data sources, such as the “CannabisGDB” data base [91], could become crucial.

The benefits of applying these approaches not just to cannabis, but to many other organisms, provide a need to rapidly overcome the current technological limitations. As such, we anticipate that in the near future, these types of studies will become far more widespread, and the full utilisation of these complex datasets will become an attainable goal for many researchers.

With changes in legislation and public attitude, the demand for research activity in cannabis is growing rapidly and new, innovative applications are developing. For instance, data provided by omics experiments are being used in parallel with genetic engineering approaches to obtain a better understanding of Cannabis genetics [92]. However, such methods are made more challenging by the fact that cannabis is recalcitrant to traditional genetic transformation methods [2] and *Agrobacterium tumefaciens* [93] is less efficient for cannabis than for many other crop species [94]. This can be overcome, in part, by employing biolistic genetic engineering approaches or transient gene transformations [94].

Furthermore, other innovative strategies that investigated the potential of a pan-genomic approach were carried out in recent years, considering the inadequacy of a single reference genome to represent the complete genetic diversity of a given plant species [95]. For instance, independent cannabis pangenome projects promoted by NRGene and Medicinal Genomics [96,97] have been carried out, and conserved genomic regions, as well as variable regions related to structural variants (SVs), CNVs and presence/absence variations (PAVs) useful in cannabis breeding were also better identified.

These emerging technologies, based on genetic engineering and/or pan-genomics approaches, and those combined with data from multi-omics applications will be able to complement each other, to rapidly overcome the current gaps in cannabis research and fully exploit the data generated to date. For instance, the role and the involvement of many candidate genes in cannabis molecular mechanisms, which is not still fully elucidated [75,80], could be better defined by using genetic engineering technologies. These could also contribute to elucidate the biological controls and mechanisms behind the enhancing of cannabinoid expression, that to date are not completely clear [63]. Furthermore, studies on the relationship between cannabinoid synthase CNVs and cannabinoid content (es. [86]) could be further explored using the latest advances in the pan-genomic field, and to apply the results in cannabis breeding programs. Pan-genomic and multi-omics approaches could also provide further clues about the nature of species diversity, already addressed in numerous studies (es. [15]) but still not entirely elucidated.

Finally, we believe that integrating well-established multi-omics approaches, such as those based on metabolomics, genomics, and transcriptomics, with other omics sciences such as phenomics, which analyses qualitative and quantitative traits for the characterization of a given phenotype [98], will greatly contribute to improve our knowledge and understanding of biological pathways. This will especially assist in elucidating those complex cannabis genotype traits that, despite the advances of high-throughput technologies, are still not well characterized from the genetic standpoint.

## Figures and Tables

**Figure 1 plants-11-02182-f001:**
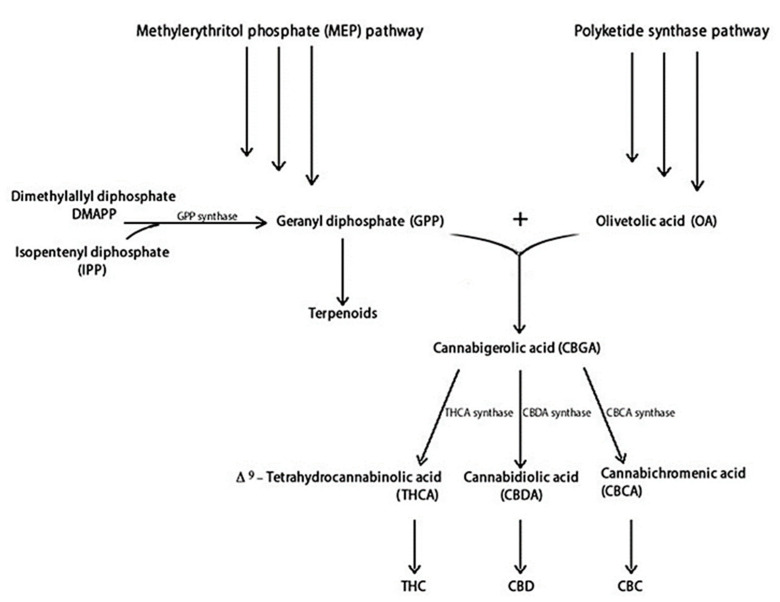
Cannabinoids and terpenoids biosynthesis in *C. sativa* [38,41].

**Table 1 plants-11-02182-t001:** List of abbreviations used in this manuscript.

Abbreviations	Definition
CBC	Cannabichromene
CBCA	Cannabichromenic Acid
CBD	Cannabidiol
CBDA	Acidic Precursors of Cannabidiol
CBG	Cannabigerol
CBGA	Acidic Precursors of Cannabigerolic
CBDRx	High Cannabidiol Content Cannabis Cultivar
CNV	Copy Number Variation
DEG	Differentially Expressed Gene
DLLLME	Dispersive Liquid-Liquid Microextraction
DMAPP	Dimethylallyl Diphosphate
FN	Finola Cannabis Cultivar
GBS	Genotype By Sequencing
GC-EIMS	Gas Chromatography-Electron Impact Mass Spectrometry
GC-FID	Gas Chromatography-Flame Ionization Detection
GC-MS	Gas Chromatography–Mass Spectrometry
GPP	Geranyl Diphosphate
GS	Genomic Selection
GWAS	Genome-Wide Association Study
HPLC	High Performance Liquid Chromatography
HRMS	High-Resolution Mass Spectrometry
HS-SPME	Headspace Solid-Phase Micro Extraction
IPP	Isopentenyl Diphosphate
ISO-Seq	Isoform Sequencing
JL	Jamaican Lion Cannabis Cultivar
LC-DAD	Liquid Chromatography-Diode Array Detector
LC-MS	Liquid Chromatography–Mass Spectrometry
MEP	Methylerythritol Phosphate
MS	Mass Spectrometry
NGS	Next Generation Sequencing
NMR	Nuclear Magnetic Resonance
OA	Olivetolic Acid
ORF	Open Reading Frames
PAVs	Presence/Absence Variations
PPFD	Photosynthetic Photon Flux Density
PK	Purple Kush Cannabis Cultivar
QTL	Quantitative Trait Loci
QQQ	Triple-Quadrupole Mass Spectrometry
qRT-PCR	Real time/Quantitative PCR—Polymerase Chain Reaction
RNA-Seq	RNA-Sequencing
ROS	Reactive Oxygen Species
SFE	Supercritical Fluid Extraction
SMRT	Single-Molecule Real-Time Sequencing
SNPs	Single Nucleotide Polymorphisms
SPE	Solid Phase Extraction
SPME	Solid Phase Micro Extraction
SVs	Structural Variants
THC	Tetrahydrocannabinol
THCA	Acidic Precursors of Tetrahydrocannabinol
WGCNA	Weighted Gene Co-expression Network Analysis
WGS	Whole Genome Shotgun Sequencing

**Table 2 plants-11-02182-t002:** Significant omics studies examined in this review.

Omic Technologies	Description	Reference
Metabolomics/GC-Cannabinoid synthase genotyping, linkage mapping and QTL analysis	Study about cannabinoids and terpenoids biosynthesis in cannabis	[38,62]
Metabolomics/Analytical methods available for cannabinoids analysis	Review—Cannabis metabolites	[32]
Metabolomics/GC-FID method	Definition of cannabis chemovars based on their terpenoid profile. The Effect of Light Spectrum on the cannabis morphology.	[46,50]
Metabolomics/Supercritical CO_2_ extraction of the cannabis inflorescence	Study of cannabinoids and terpenoids biosynthesis in cannabis	[40]
Metabolomics/GC-MS/LC-MS	Overview about methods for the chemical characterization of cannabis	[44]
Metabolomics/HRMS—LC-HRMS	Potential in the definition of cannabis chemovars of HRMS techniques	[60,61]
Metabolomics/SFE/SPE	Isolation of tetrahydrocannabinol from cannabis	[52]
Metabolomics/SPME/DLLME/LC-QQQ-MS	Physicochemical characterization of cannabis	[53]
Metabolomics/Analytical methods available for cannabinoids and terpenoids analysis	Review—Cannabinomics: Metabolomics applications in Cannabis. Terpenoids properties.	[20,42]
Metabolomics/Structural classification of phytocannabinoids	Study about phytocannabinoids	[39]
Metabolomics/GC-EIMS	Investigation about lipids extracted from cannabis seeds	[47]
Metabolomics/HPLC	Investigation about the major cannabinoids: CBD, CBG and THC.	[48,49]
Genomics/Illumina sequencing approach	The genome of cannabis. To assess the completeness and representivity of the ‘PK’, ‘FN’, and ‘CBDRx’ assemblies, Illumina sequences and 55 public whole-genome-sequenced samples were used	[5]
Genomics/Reference assemblies were annotated with mRNA sequencing (Iso-Seq) approach	Sequence and annotation of 42 cannabis genomes	[25]
Genomics/Draft genome sequence using PacBio single-molecule sequencing	A reference genome of wild cannabis	[27]
Genomics/PCR genotyping/SNP markers	Phylogenetic relationship, population genetics, and correlation with cannabis metabolites	[8]
Genomics/GWAS	Investigation of novel cannabis genetic variants responsible for cannabis complex traits	[68,69]
Genomics/whole-genome resequencing approach	A whole-genome resequencing of of 110 worldwide accessions	[64]
Transcriptomics/RNA-Seq approach	Study of the cannabis transcriptome. Transcriptomic applications investigated fibers cannabis development stages. Cannabis salt-responsive genes were also investigated	[24,77,79,80]
Transcriptomics/De novo transcriptome assembly pipeline and BLAST2GO tool	Definition of hight quality reference transcriptomes of two cannabis cultivars	[65]

**Table 3 plants-11-02182-t003:** The multi-omic studies examined in this review.

Omic Technologies	Description	Reference
Metabolomics/GC, Transcriptomics/RNA-Seq e PCR	Quantification of the transcript levels of different cannabinoid synthase genes	[72,83,85]
Metabolomics/HPLC, Genomics/WGS and GBS	Investigation about cannabis ancestry	[15]
Metabolomics/SPME GC-MS and LC-MS, Transcriptomics/Rna-Seq	Investigation about trichomes changes during flower maturation in cannabis	[33]
Metabolomics/HPLC/GC, Transcriptomics/RNA-Seq	Study about nine cannabis cultivars having different basic characteristics	[43]
Metabolomics/HPLC, genomics/WGS, Transcriptomics/RNA-Seq	Study about the relation between CNVs and cannabinoids profile	[86]
Genomics/SNPs, Transcriptomics/RNA-sequencing pipeline based on CG technology	Investigation about sex chromosome evolution in cannabis	[67]
Genomics/PacBio, Transcriptomics/RNA sequencing based on Iso-Seq	Study about the cannabis sex evolution and the pathogen resistance	[65]
Genomics/Transcriptomics—Innovative approaches discussion, including GWAS, GS, pan-genomics	Review about cannabis genomics resources	[19]
Metabolomics, genomics and transcriptomics data are collected and structured, and available on a web site for researchers	“CannabisGDB”, a comprehensive multi-omics database	[91]
Metabolomics, Genomics and transcriptomics platforms. WGS approach in medical cannabis is discussed	Review on medicinal plants multi-omics applications	[82]
Metabolomics/GC, Genomics/WGS/Linkage mapping and QTL analysis	Comparation between Carmagnola and USO31 cannabis cultivars	[89]
Metabolomics/HPLC, Genomics/Nanopore technology/Linkage mapping and QTL analysis, Transcriptomics/RNA-Seq	Study about the CBDRx cannabis cultivar	[63]

## Data Availability

Not applicable.

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
