# Peer review of "Multi-Omics Approaches to Study Molecular Mechanisms in Cannabis sativa"

_plants, 2022, doi:10.3390/plants11162182_

Round 1

Reviewer 1 Report

In table 1, authors mentioned GWAS. But there are no sentences about GWAS in the paper. GWAS is an important tool in genetic dissection. It is better to add it to the paper.

Genomic selection using multi-omics seems to be a good choice in breeding. I know some papers about it. It is better to add some content about genomic selection using multi-omics. The papers below should be cited.

Xu, S., Xu, Y., Gong, L., & Zhang, Q. (2016). Metabolomic prediction of yield in hybrid rice. The Plant Journal88(2), 219-227.

Wang, S., Wei, J., Li, R., Qu, H., Chater, J. M., Ma, R., ... & Jia, Z. (2019). Identification of optimal prediction models using multi-omic data for selecting hybrid rice. Heredity123(3), 395-406.

Wang, S., Xu, Y., Qu, H., Cui, Y., Li, R., Chater, J. M., ... & Jia, Z. (2021). Boosting predictabilities of agronomic traits in rice using bivariate genomic selection. Briefings in Bioinformatics22(3), bbaa103.

 Knoch, D., Werner, C. R., Meyer, R. C., Riewe, D., Abbadi, A., Lücke, S., ... & Altmann, T. (2021). Multi-omics-based prediction of hybrid performance in canola. Theoretical and Applied Genetics134(4), 1147-1165.

Reviewer 2 Report

Cannabis is promising for biomass crop as fiber as well as chemical properties

Due to the legislation regulating cannabis cultivation, many of characteristics have not well identified.   This review paper describes a summary of cannabis molecular resources focusing on the most recent and relevant genomics, transcriptomics and metabolomics approaches and investigations.   It is well worth publishing because it contains important information on breeding and other important information in the future.

My major suggestion as follow:

1.     The authors simply listed the omics studies in both Table 2 and Table 3.

It is better to have a list of information about each omics technology, rather than a list of scientific papers.

2.     I feel that there is not much information on the plant phenotype, although I don't think there is much to study.

My minor suggestion as follow:

Introduction

   Botanical basic information such as reproductive characteristics, genome composition and so on could be necessary in the first paragraph.   

Scientific name nomenclator letter is normal font.  Also some scientific name is nothing for nomenclator.

Is the Table I inserted in the proper position?

It would be better to use ' … ' for the description of name of the variety
